Medium-term acoustic monitoring of small cetaceans in Patagonia, Chile

Patris Julie julie.patris@univ-amu.fr 1
Malige Franck 2
Hamame Madeleine 3
Glotin Hervé 2 4
Barchasz Valentin 4
Gies Valentin 4 5
Marzetti Sebastian 4 5
Buchan Susannah 6 7 8
1 Université d’Aix-Marseille , Marseille , France
2 Laboratoire Informatique et Systèmes (LIS), CNRS UMR 7020 , Toulon , France
3 Centro de Investigación en Ecosistemas de la Patagonia (CIEP) , Coyhaique , Chile
4 Scientific Microsystems for Internet of Things (SMIoT), Université de Toulon et du Var , Toulon , France
5 Institut Matériaux Microélectronique Nanosciences de Provence (IM2NP), CNRS UMR 7334 , Toulon , France
6 Center for Oceanographic Research COPAS COASTAL, Universidad de Concepción , Concepción , Chile
7 Departamento de Oceanografía, Universidad de Concepción , Concepción , Chile
8 Centro de Estudios Avanzado en Zonas Aridas (CEAZA) , Coquimbo , Chile
Gandini Patricia
Electronic publication date: 2023 Jun 14
Publication date: 2023
Volume: 11
Electronic Location ID: e15292
Received 2022 Jan 6; Accepted 2023 Apr 4
Copyright: ©2023 Patris et al.
Copyright year: 2023
Copyright holder: Patris et al.
License: This is an open access article distributed under the terms of the Creative Commons Attribution License, which permits unrestricted use, distribution, reproduction and adaptation in any medium and for any purpose provided that it is properly attributed. For attribution, the original author(s), title, publication source (PeerJ) and either DOI or URL of the article must be cited.
License URL: https://creativecommons.org/licenses/by/4.0/

Keywords: Bioacoustics, Remote sensing, Patagonian ecosystem, Coastal small cetaceans, Chilean dolphin, C-POD

Funding: The Programa Regional ANID of Chile R17A10002 R20F0002 This work was funded by the Programa Regional ANID of Chile (No. R17A10002 and R20F0002). The funders had no role in study design, data collection and analysis, decision to publish, or preparation of the manuscript.

==============================
Coastal dolphins and porpoises such as the Chilean dolphin (Cephalorhynchus eutropia), the Peale’s dolphin (Lagenorhynchus australis), and the Burmeister’s porpoise (Phocoena spinipinnis) inhabit the remote areas of Chilean Patagonia. Human development is growing fast in these parts and may constitute a serious threat to such poorly known species. It is thus urgent to develop new tools to try and study these cryptic species and find out more about their behavior, population levels, and habits. These odontocetes emit narrow-band high-frequency (NBHF) clicks and efforts have been made to characterize precisely their acoustic production. Passive acoustic monitoring is a common way to study these animals. Nevertheless, as the signal frequency is usually higher than 100 kHz, storage problems are acute and do not allow for long-term monitoring. The solutions for recording NBHF clicks are usually twofold: either short duration, opportunistic recording from a small boat in presence of the animals (short-term monitoring) or long-term monitoring using devices including a click detector and registering events rather than sound. We suggest, as another possibility, medium-term monitoring, arguing that today’s devices have reached a level of performance allowing for a few days of continual recording even at these extremely high frequencies and in difficult conditions, combined with a long-term click detector. As an example, during 2021, we performed a quasi-continuous recording for one week with the Qualilife High-Blue recorder anchored in a fjord near Puerto Cisnes, Region de Aysen, Chile. We detected more than 13,000 clicks, grouped in 22 periods of passing animals. Our detected clicks are quite similar to precedent results but, due to the large number of clicks recorded, we find a larger variability of parameters. Several rapid sequences of clicks (buzz) were found in the recordings and their features are consistent with previous studies: on average they have a larger bandwidth and a lower peak frequency than the usual clicks. We also installed in the same place a click detector (C-POD) and the two devices compare well and show the same number and duration of periods of animals presence. Passages of odontocetes were happening on average each three hours. We thus confirm the high site fidelity for the species of dolphins emitting NBHF clicks present in this zone. Finally, we confirm that the combined use of recording and detection devices is probably a good alternative to study these poorly known species in remote areas.

Introduction

Coastal small odontocetes are present in many zones of the world, including rivers, fjords and bays. Due to their site fidelity they usually are very sensitive to human presence and some populations are on the verge of extinction (Jaramillo Legorreta et al., 2019; Sucunza et al., 2019; Silva et al., 2020). Many studies of these species focused on areas where human activity and presence is high, because it is usually easier to reach these areas and because the threats are stronger (Heinrich et al., 2019; Palmer et al., 2021). In remote areas such as Patagonia, there is still little information available on these species, though they are probably also threatened and population assessments could be decisive for their conservation.

Long term visual studies are costly and are subject to the contingencies of climate and to the locally available equipment (Stern et al., 2017; Heinrich et al., 2019). Passive acoustic monitoring (PAM) is sometimes a good alternative to assess the presence, sound characteristics and behavior of cetaceans, or to estimate their density and population trends (Marques et al., 2013), especially in remote areas (see for example Schall et al. (2021)). However, in the case of odontocetes emitting narrow band high frequency (NBHF) clicks, there is a serious drawback to PAM methods: the high sample rate needed to record their high frequency emissions prevents autonomous long term recording. The very few published studies that used long term recording had very expensive devices and installations, that are not commonly found in marine biology (Gillespie et al., 2020). Usually, there are two alternatives for the passive acoustic monitoring of small coastal cetaceans: short term recording or long term presence detection.

The first method consists in recording during short time periods typically a few hours or less, usually opportunistically from a boat in the wild or in a pool for captive animals. The recording is controlled, sometimes with several hydrophones (array of sensors) and the behavior of the animal is registered (Ladegaard et al., 2015; Macaulay et al., 2020; Barlow, Cheeseman & Trickey, 2021). This kind of work is useful for describing the emissions in details (sound characteristics, beam), and/or coupling them with behavioral observations. Nevertheless, as these studies are short in duration or done in captivity, the presence of humans is a possible source of disturbance that can affect the behavior and sound production of these marine mammals. Thus, this type of study is mainly focused on characterizing the sounds emitted by a particular species, but could be biased towards certain types of sound emissions or conducts in reaction with human presence such as anxious, agonistic, attentive, or cautious behaviors (Martin et al., 2021).

The second widely used method is long term monitoring with click detectors (Sousa-Lima et al., 2013; Weel et al., 2018). Click detectors do not fully record the signal, but detect and log predetermined sounds of interest along with some of their characteristics. Thus, memory use and power consumption are much lower than for recorders, and an area can be monitored for years, due to the high autonomy of the available detectors. A drawback of these very efficient tools is that very little information is then available on the surrounding low to medium frequency sounds or soundscape. For instance, detectors can hardly be used to assess interactions between marine mammals and human produced noises. Moreover, the differentiation of sounds emitted by species of interest by a logging device is not easy (Jacobson et al., 2017), particularly in species whose repertoire is similar or still not fully known. In addition, the calibration of such devices is often a problem since the data is not recorded and no a posteriori verification can be done (Robbins et al., 2015). To solve this problem some studies proposed a combination of a long term detector and a recording device, used for calibration purpose, mainly to test the detector performance (Jacobson et al., 2017; Sarnocinska et al., 2016). Interestingly, instruments combining low frequency recording, automatic detection and high frequency snippet recording will soon become available (http://www.oceaninstruments.co.nz/) though no studies using them have been published yet, to our knowledge. This is an exciting new technology, even if the reliability of the detector is still a potential difficulty.

In this work, we suggest, as another possibility, a medium-term full recording monitoring for small coastal cetaceans along with a long term monitoring by mean of a click detector. We argue that today’s recording devices have reached a level in performance allowing for a few days of continual recording even at these extremely high frequencies and in difficult conditions or remote places. Custom-built recorders allow for adaptation to special conditions or a specific protocol at a relatively low cost. This set-up of two joint devices combines several qualities : the medium-term recording provides comprehensive information on the sound produced by these coastal odontocetes (and enables future studies in signal processing), of the acoustic context (noises, human and other animal sound emissions), can help to calibrate the logging of predetermined sounds by the detector and is less invasive compared to other approaches such as recording from a boat. We present an example of such a medium-term recording in the remote fjords of Chilean Patagonia in May 2021, aiming at testing the feasibility of such a monitoring approach as well as knowing better the acoustical repertoire of the cryptic small cetaceans inhabiting the inlet waters. After presenting the species of interests, we describe in detail the two instruments used, compare the detections of each device, show some biological results that can be obtained and discuss the advantages and disadvantages of this experimental set up in remote places.

Coastal odontocetes in Patagonia

Fjords of northern Chilean Patagonia

The marine ecosystem of Chilean Patagonia (41°5′–55°S) is considered one of the most extensive fjord systems in the world. Numerous islands, peninsulas, channels, straits and fjords form part of its complex geography covering an area of ca. 240 000 km2 (Silva & Vargas, 2014). Oceanographically, sub-antarctic water, rich in nutrients, flow on the surface through “Boca del Guafo” (43°35.7′S–74°12.8′W) mixing progressively towards the south with estuarine water (Guzmán & Silva, 2006; Silva & Palma, 2008). This oceanographic and geomorphological particularities create many unique habitats that result in a high degree of endemic wildlife and high species richness (Häussermann & Försterra, 2009; Försterra, Häussermann & Laudien, 2017; Betti et al., 2017). The region is classified as highly vulnerable to local and remote processes (Iriarte, González & Nahuelhual, 2010). Major threats associated to economic activities includes intense salmon farming, demersal and benthic artisanal fisheries and emerging cetacean sightseeing activities.

Small coastal cetaceans of northern Chilean Patagonia

Chile is among the countries with the larger diversity of cetaceans, mainly due to its large coastline and variety of climates (Wilson & Mittermeier, 2014). The remote fjords and inlet waters of Aysén (Northern Chilean Patagonia) are no exception to this diversity (Zamorano-Abramson, Gibbons & Capella, 2010; Pichinao et al., 2019). Large delphinids, such as the bottlenose dolphin (Tursiops truncatus) or the predating killer whale (Orcinus orca) are transient regular visitors of the fjords, and large mysticetes such as the blue whale (Balaenoptera musculus), the Sei whale (Balaenoptera borealis) or the humpback whale (Megaptera novaeangliae) are common in the larger channels. Inside the fjords however, and very close to the shore, three species of small cetaceans mostly share the sheltered habitat: the Burmeister’s porpoise (Phocoena spinipinnis), the Peale’s dolphin (Lagenorhynchus australis) and the Chilean dolphin (Cephalorhynchus eutropia).

These three species are endemic to South America, the Chilean dolphin being even restricted to Southern Chile. They are globally poorly known, with very few studies published, and especially in the inlet waters of Chilean Patagonia. Their conservation status is considered ‘near threatened’ for the Chilean dolphin (Heinrich & Reeves, 2017) (mainly because of their restricted ranges) and the Burmeister’s porpoise (Félix et al., 2018) (because of bycatch threats). Human activities in coastal areas are generally a major threat to coastal cetaceans, through interactions with gill nets, fisheries or farms (Félix et al., 2018; Heinrich et al., 2019). The Peale’s dolphin is often seen in the fjords porpoising around the boats or foraging close to the shore. The Burmeister’s porpoise and the Chilean dolphin are much more elusive, and do not normally interact with the boats.

In the Chilean fjords of Patagonia, only these three species of coastal odontocetes emit NBHF echolocation clicks. Interestingly, for each species, only one study describing their vocalization has been published (Reyes Reyes et al., 2018; Kyhn et al., 2010; Götz, Antunes & Heinrich, 2010). Additionally, one study compared the emitted signals of Chilean and Peale’s dolphins (Rojas-Mena, 2009). The NBHF click is common in coastal species of toothed whales (Galatius et al., 2019). It is characterized by a peak frequency around 130 kHz, a half-power bandwidth of about 15 kHz and almost no energy below 100 kHz. It is thought to be an adaptative response to the predation of killer whales, that do not hear above 100 kHz (Andersen & Amundin, 1976; Morisaka & Connor, 2007). Recent studies point out that some species of the Cephalorhynchus genus can relax this acoustic crypsis, emitting clicks at lower frequencies probably in a communication context (Martin et al., 2018; Martin et al., 2021). In addition, the three species have been shown to produce ‘buzz’, or very rapid trains of clicks thought to be used while foraging (Götz, Antunes & Heinrich, 2010; Martin et al., 2019; Rojas-Mena, 2009; Reyes Reyes et al., 2018). NBHF signals are very similar between species, and are possibly depending on the environment more than on the species (Kyhn et al., 2010), hence the need for more studies on these species vocalizations, that could allow for future long term passive acoustics monitoring by means of accurate detectors.

An experiment in the fjord of Puyuhuapi

Material and methods

QHB Recorder

The main instrument for the experiment is a Qualilife HighBlue (QHB) recorder, developed by SMIoT, University of Toulon, and presented in Fig. 1.

Its functional diagram is shown in Fig. 2. This recorder has the following characteristics (see also Barchasz et al. (2020)):

• Acquisition sample rates up to 512 Ksps (Kilo samples per second) corresponding to a frequency range up to 256 kHz. Recording can be scheduled according to user requirements.

• Up to six synchronous recording channels, with an accurate synchronization and time-stamping having less than 1 µs of jitter.

• Signal sampling depth can be adjusted among 8, 16 or 24 bits. In this latter mode, recorder self noise is limited to the two least significant bits, meaning 22 bits are truly significant for recording. This increases the signal quality and the potential detection distance compared to standard recorders, especially in quiet environments.

• Differential acquisition front end with ±2.5V maximum input level for reducing drastically recording self noise. Each recording channel has an adjustable differential gain: X1, X10, X20, X100.

• Anti-aliasing filtering automatically tuned according to the acquisition sampling rate. Signal having frequencies exceeding 0.55* Sampling Rate are attenuated by more than 120 dB.

• Sensor hub ability: QHB includes a 9-axis IMU sensor (MEMS accelerometer, magnetometer and gyroscope) and several additional sensors can be added depending on user requirements, using UART, SPI and I2C extension buses.

The QHB recorder was set up in a custom made housing allowing resistance to pressure up to 100 m deep, a stable setting on the ground, the adaptation of a C57 hydrophone from Cetacean Research, calibrated with a flat response ± 3 dB up to 250 kHz (calibration done in the Laboratoire de Mécanique et d’Acoustique, Marseilles), and a set of 21 D alkaline batteries. A reference for the QHB recorder can be found at: http://bioacoustics.lis-lab.fr/smiot.

Figure 1 Qualilife HighBlue (QHB) recorder.

Figure 2 Functional diagram of Qualilife HighBlue (QHB) recorders.

C-POD

Though the main instrument of the experiment was the QHB recorder, we also installed a C-POD, a commercial click detector developed by Chelonia Limited, UK (Tregenza, 2014). The C-POD works in the 20 kHz–160 kHz range, detects and logs all potential clicks in this frequency range, registering several parameters for each detection (central frequency, duration, etc.) as well as the ambient sea water temperature. A post-processing software classifies the detections between high frequency noise and real clicks based on the properties of the train of clicks, further offering a classification between NBHF or medium frequency (odontocetes) click. The C-POD is widely used for long term monitoring of toothed whales, and especially the harbour porpoise (Phocoena phocoena) because of its low energy consumption, low memory requisite and hence its very large autonomy in the field (Sousa-Lima et al., 2013; Gallus et al., 2012).

Data recording

Both instruments QHB and C-POD were set on May, 4th of 2021, in a cove close to the shore of Magdalena Island reserve, in the canal of Puyuhuapi opposite the town of Puerto Cisnes (44°36′38.78″S, 72°45′30.43″W, Fig. 3).

The place was chosen because local tour operators had seen repeatedly Chilean dolphins in this cove during the last months. The instrument QHB was installed at a depth of approximately 13 m, on sandy ground (Fig. 3). At 10 m of distance, a mooring was set with a line sustaining the C-POD (at 4m from the ground) and a subsurface buoy. The set up of QHB was a sample rate of 512 kHz, 24 bits of precision, one channel, and a duty cycle of 95% with 9’30” of recording followed by 30” OFF. The C-POD was used with default settings: continuous logging and a 20 kHz high-pass filter (Tregenza, 2014). The QHB was retrieved on May, 11th whereas the C-POD was retrieved on July, 28th. Only Chilean dolphins were observed inside this cove, either by the authors or by the tour operators visiting the place. The only moment when we saw the dolphins was during the operation of changing the memory card on the 8th of May, when two individuals of a group of about 15 Chilean dolphins stayed with the diver, interacting below the water.

Click detection

A click detector was custom written in Octave (Eaton, Bateman & Hauberg, 2009). It basically detects the maxima of energy in the frequency band of 100 kHz–250 kHz, and then filters out the signal that have a strong counterpart in the 30–90 kHz bandwidth. Our detector was tested on two 9.5 minutes long files, with clicks identified by a human specialist. The first file had a lot of clicks (N = 523) and some high frequency noise, and the other file was without detected click but with a lot of high frequency noise. For the chosen thresholds, we determined the following characteristics:

Figure 3 (A) Study location in South America. (B) Zoomed in view on Canal Puyuhuapi. In blue, the point chosen for the installation of the different devices (44°36′38.78″S, 72°45′30.43″W). (C) Mooring design for both the QHB recorder and the C-POD.

• Precision or positive predicted value (PPV = correctly detected / all detections) PPV = 84%

• Miss rate (MR = missed signals / all signals) MR = 17% .

These are conservative values since we chose, for the testing subset, two of the noisiest files. The code for this simple detector is given as supplementary material.

Extraction of clicks parameters

As a first analysis of the clicks, we wrote a short code to automatically extract the most commonly used parameters of NBHF clicks (Au, 1993), in concordance with the papers published about the NBHF clicks of the three species of odontocetes present in the Fjord of Puyuhuapi (Götz, Antunes & Heinrich, 2010; Kyhn et al., 2010; Reyes Reyes et al., 2018). The code is given as supplementary material and it computes the classical parameters listed below. Peak frequency is computed as the maximum of the Fast Fourier Transform(FFT) of 512 samples (1 ms) around the clicks. Centroid frequency (or mean frequency) is the first raw moment of the FFT of the recorded signal during the same extract. Inter-click interval (ICI) is computed as the time between two detections closer than 300 ms. In the (infrequent) case of two superimposed trains of clicks, this measure does not reflect an intrinsic property of the emitted sound. Frequency bandwidth RMS (Root Mean Square) is the second central moment of the distribution of frequencies in the same 1 ms extract. Bandwidth at -3 dB is the frequency band around the peak frequency where the value of the Fast Fourier Transform (FFT) is higher than the maximum of the FFT divided by 2. Bandwidth at -10 dB is the frequency band around the peak frequency where the value of the Fast Fourier Transform (FFT) is higher than the maximum of the FFT divided by 10. RMS duration is the second central moment of the distribution of time, where the modulus squared of the signal divided by its energy is considered a probability density function. Duration at -10 dB is the duration around the maximum of the signal where the envelope of the signal is higher than the maximum of the signal divided by 10. The enveloppe is obtained using the absolute values of the analytical signal (Hilbert transform in Octave) corresponding to 1 ms of real-valued signal around the peak of the clicks. Duration at -20 dB is the duration around the maximum of the signal where the envelope of the signal is higher than the maximum of the signal divided by 10. The statistical distribution of each of these parameters is then computed for all the data set.

The clicks are organized in trains of several clicks and usually grouped in ‘events’ or encounters. We defined an ‘event’ as a series of trains separated by less than 20 min. This definition is due to the observation that the number of ‘events’ obtained is less variable for this time scale.

Results

Clicks and events detections

The QHB instrument recorded well from the 4/05/21 at 11h30 local time to the 6/05/21 at 20 h local time, when it had a failure: it began to record the sound on the same file of 9’30” for the rest of the recording session. Then, in the second session, it recorded from the 8/05/21 at 11 h to the 10/05/21 at 11 h local time until it ran out of battery. We thus have two periods of recording, one of 56 h with 339 files of 9’30” and one of 48 h with 291 files of 9’30”. We total more than 550 GB of recorded sound.

We detected more than 13,000 clicks during the 56 h from the 4th to the 6th of May, and almost none in the second period from the 8th to the 10th of May. With the definition of ‘event’ presented in the previous section, we find 22 events or encounters during the 56 h. Events were separated by intervals from 30 min to 6.5 h (Fig. 4). The C-POD detector recorded from the 4/05/21 to the 27/07/21. Although all the data have been extracted from the instrument, amounting to about 34,000 clicks (all classified as NBHF) during the whole three months, only the period when both instruments were in the water has been analyzed here. Figure 4 shows the compatibility of the results between the QHB instruments and the C-POD detector for the first three days, when a lot of clicks were detected by both instruments. Most of the events were detected by both the instruments, even though they were about 10 m apart. However, the detection rate of the QHB is significantly higher (more than 13,000 clicks as opposed to about 2,000 clicks for the C-POD for the same period). The number of chunks of 10 min with at least one detection is 38 in total for the CPOD and 49 for QHB.

Figure 4 Number of clicks detected per 10 min by QHB (blue) and the C-POD (purple).

Twenty of the 22 events detected by the QHB recorder have also been detected by the C-POD instrument. Superimposed are night and day (night in grey, day in white) and tides in arbitrary units (black curve).

QHB instrument also recorded contextual noise such as boat engines and sonars, as well as long duration motor noise probably linked to a nearby salmon farm (situated at about 2 km), and noise from the natural environment such as crabs, shrimps, etc. However, no detailed analysis of background noise has yet been done. It is intriguing to note that in both instruments, no clicks were detected between the 8/05 in the morning (when we changed the memory card, with two Chilean dolphins interacting with the diver) and the 10/05 late at night. On the 11th of May, the QHB instrument was removed. In the data of the C-POD, such large intervals without click are quite unusual (only three registered in the three months of data).

Clicks properties

The clicks that were registered by QHB have a good definition and are similar to the NBHF clicks described in the literature (Rojas-Mena, 2009; Götz, Antunes & Heinrich, 2010; Kyhn et al., 2010; Reyes Reyes et al., 2018). The clicks’ average parameters are given in Table 1.

Table 1 Parameters of the clicks recorded by QHB instrument (average value and standard deviation, N = 13,878).

Peak frequency	Frequency bandwidth ‘rms’	Duration ‘rms’	
(135 ± 15) kHz	(19 ± 5) kHz	(57 ± 21) µs	
Centroid frequency	Frequency bandwidth at -3 dB	Duration at -10 dB	
(141 ± 10) kHz	(6 ± 3) kHz	(53 ± 26) µs	
Inter-click interval (ICI)	Frequency bandwidth at -10 dB	Duration at -20 dB	
(88  ± 117) ms	(16 ± 8) kHz	(106 ± 52) µs	

Nevertheless, the statistical distributions of the parameters are not all Gaussian, as can be seen in Fig. 5. This is particularly the case with the distribution of ICI, with a standard deviation larger than the average value and two very different modes in the distribution, and the peak frequency distribution, which is clearly multi-modal. Therefore, the description of the parameters using average values and standard deviations is not the best way to describe the diversity of clicks recorded.

Figure 5 (A–D) Distributions of the parameters of the detected clicks (N = 13,878).

Average and standard deviation are given in Table 1.

The main peak of the distribution of peak frequency is itself bi-modal with a mode around 126 kHz, and another at 134 kHz. On the other hand, a mode is visible at very high frequency around 164 kHz. Three examples of clicks are given in Fig. 6. We found that some of the clicks had a large bandwidth, with some having a peak of energy at 170 kHz. A clear notch is also present in the spectra at 150 kHz as noticed by Reyes Reyes et al. (2015) for the Commerson’s dolphin. Interestingly, this notch at around 150 kHz has also been described for different species of porpoises (Reyes Reyes et al., 2018).

Figure 6 Examples of NBHF clicks recorded by QHB.

On the left (A-D-G), a typical click with peak frequency around 135 kHz. In the center (B-E-H), a less commonly recorded click with peak frequency around 180 kHz. On the right (C-F-I) an example of a click found in a buzz, or rapid sequence of clicks. Top: spectrogram of the signal with a FFT on 210 points (2 ms) except for the right picture (27 points, 0.5 ms), Blackman window, 50% overlap. Middle: zoom on the waveform of the click at the center of the figure just above. Bottom: spectra of the click with normalized intensity, FFT of 29 = 512 points (1 ms of the signal), centered on the detection.

Also present in our data set are very rapid trains of clicks, usually denominated buzz (Figs. 6C, 6F, 6I). We define a ‘buzz’ when the ICI is lower than 5 ms, usually around 2 ms, as compared to normal trains with ICI being between 50 and 100 ms. A visual examination of our data show about 20 such trains, seven of them within the same file of 9’30”. These clicks are visible in the ICI distribution (very short ICI, Fig. 5G). Visual examination of the clicks with short ICI confirmed there was no superimposed trains of clicks, and thus the ICI actually corresponded to an intrinsic parameter of the emitted sound. The last mode in the distribution of frequency peak, around 107 kHz, corresponded to these buzzes. It is coherent with the results of Götz, Antunes & Heinrich (2010), Reyes Reyes et al. (2015) and Martin et al. (2021) that NBHF click species emit buzz clicks at a slightly lower frequency. The clicks found in a buzz, or rapid sequence, have clearly different features compared to other clicks. The number of cycles included in the envelope of the click is much lower than for classic NBHF clicks, and shows some similarity with typical clicks of larger odontocetes. The spectrum shows a greater bandwidth, with energy lower than 75 kHz. Though we had no means of measuring the distance of the animals to the sensor, and thus we could not calculate source levels in this study, the buzz clicks that we found are generally of lower intensity compared to nearby classic NBHF clicks.

Discussion

Validation of C-POD detections

Our results concerning the comparison between C-POD detectors and a recording device are twofold. On the one hand, the absolute numbers of detections are widely different between the two instruments. On the other hand however, almost all ‘events’ have been detected by both. Although this comparison between C-POD detector and full signal recording has never been done for NBHF clicks emitted by the three species presented in this study, it has been measured for other species, such as the harbor porpoise, one of the species most studied with clicks detector, with somewhat distinct conclusions. While Sarnocinska et al. (2016) found a rather low correlation between the clicks per minute detected by a C-POD detector and a Soundtrap recording device, installed at a distance of about 2 m in the same mooring line, Jacobson et al. (2017) found a much better correlation between the results of the same two instruments, installed so that the two hydrophones were as close as possible. Such differences may be due to the respective position of the instruments, but, more importantly, by the difference of sensitivity of each individual instrument. In our experiment, it is obvious that the recorder is much more sensitive than the detector, independently of the difference of the location of the instruments. However, and though the numbers of detected clicks show a difference of 600%, the number of detected ‘events’ is a much more robust indicator. Indeed, 20 of the 22 events detected by the QHB recorder have also been detected by the C-POD instrument, a difference of hardly 5% (concerning the weakest events, see Fig. 4). The two events detected by the QHB recorder and not by the C-POD contain slightly more than ten click detections. One of them contains only false positives and, in the other, there are mostly NBHF clicks. The classical parameters of chunks of ten minutes of recording with positive detection is thus much more robust to the global sensitivity of the instrument than the absolute number of detections. The size of the chunks should be defined after considering the data, since it can be very different for each experiment, depending on the size of home range of these odontocetes, the number of groups inhabiting the area, etc. In this context, our study validates the use of a C-POD device for long-term monitoring of these three species in the Patagonia fjords.

Click properties

Our data set is much larger than the pioneer works of Götz, Antunes & Heinrich (2010), Kyhn et al. (2010) and Reyes Reyes et al. (2018). Almost 14,000 clicks were analyzed in our study, as compared to around 3,000 summing these three studies. The distribution of peak frequencies in our data set shows a large diversity. Thanks to the large number of clicks of our data set, we can compute the values of the main peaks by fitting a sum of Gaussian functions on the histogram of peak frequencies of Fig. 5. Using an implemented Marquardt-Levenberg algorithm in Octave, we find that the peak frequencies are 105.8, 125.1, 135.5 and 168.3 kHz (with respective standard deviations of 4.4, 6.0, 4.4, and 18.0 kHz, see Fig. 7). Götz, Antunes & Heinrich (2010) data (N = 83 clicks) is mainly similar to our second mode (second in order of importance) at 126 kHz. The first mode at 134 kHz is compatible with productions of Burmeister’s porpoise as measured by Reyes Reyes et al. (2018).

Previous studies of the acoustic productions of the three species present in Patagonian fjords give only the average and standard deviation for the peak frequency distribution (see Reyes Reyes et al. (2018), tab2, for a summary). In the results summarized in Reyes Reyes et al. (2018), the standard deviation of peak frequency measures increases with the number of clicks indicating that several modes are possibly appearing in a richer data set.

Interestingly, the modes we found in the peak frequency distribution are similar to what Reyes Reyes et al. (2015) describe for the Commerson’s dolphin (Cephalorhynchus commersonii), a close congener of the Chilean dolphin found mainly in the Argentina coast, sub antarctic islands and Southern Chilean Patagonia (Crespo et al., 2017). This species, however, is not present in the fjords of Northern Chilean Patagonia. They describe three clusters of clicks for this species, highly similar to what we found, with the median for each cluster being respectively at 129, 137 and 173 kHz. This study of the Commerson dolphin (Reyes Reyes et al., 2015) shows a dissimilarity with the pioneer measures of Kyhn et al. (2010), with higher average frequencies and a much larger standard deviation for peak or centroid frequencies. The study by Reyes Reyes et al. (2015) analyzed a large number of clicks (as our study), which could explain the similarity of the results. Another example is given in Reyes Reyes et al. (2018), for the Burmeister’s porpoise: some of the five hundred clicks analyzed have a peak frequency around 170 kHz and a histogram with two modes was obtained. Reyes Reyes et al. (2015) study of the Commerson’s dolphin did not find low frequency (100 kHz) clicks, nor large band clicks in a buzz, though some have been described afterwards by Martin et al. (2021).

Figure 7 Fitting of the four peaks of the first histogram in Fig. 5 (peak frequencies) by a sum of four Gaussian functions.

In our study we cannot assert if the four modes histogram comes from one species (as in Reyes Reyes et al., 2015; Reyes Reyes et al., 2018) or is the result of the mixing of different clicks from several species. In the absence of visual monitoring which would confirm the species recorded, our experiment set up draws a picture of the NBHF clicks found in a particular place rather than for a particular species. Visual monitoring would be very important to assess if the NBHF clicks are from one species or more. Nevertheless, we find in our large set of data novel types of clicks that are particularly rich and interesting. The high frequency component of the clicks cannot be found by automated detectors such as C-POD (low-pass filter at 160 kHz) or widely used recorders such as classical versions of Soundtrap (low-pass filter at 150 kHz).

A possible explanation for the presence of several modes in the histogram of Fig. 7 is that the modes could come from on and off-axis clicks. Indeed, we measured the parameters of all detected clicks without trying to select only on-axis clicks. To compute the average, we also took all detected clicks, without reference to on-axis or off-axis clicks. However, we find that each series of clicks has consistent parameters. Clicks with non-standard peak frequency (such as 100 kHz or 170 kHz) come in a series. Thus, taking the highest SNR click of a train (a classical method for selecting on-axis clicks as presented by Götz, Antunes & Heinrich (2010)) would not alter our results. There is no precise study available describing the beam pattern of the three species of odontocetes present in Puyuhuapi Fjord, however, based on measurements of the NBHF clicks of harbor porpoise (Macaulay et al., 2020), we can expect a narrow beam with little deformation of the clicks in a cone of 10° and then a high attenuation (of more than 10 dB) making the detection more difficult. On the whole, Götz, Antunes & Heinrich (2010) found very little difference on the average peak frequency between ‘on-axis’ clicks and the total set. For all these reasons, we consider that the four-mode peak frequency distribution is not a consequence of a distortion of the clicks due to the angle of reception, but reflects an intrinsic diversity of emitted clicks.

Concerning the buzzes, our data does not allow a clear separation between ‘buzz’ and ‘burst pulse’ as suggested by Martin et al. (2018) for the Heaviside’s dolphin (Cephalorhynchus heavisidii). While some of the rapid trains are part of normal trains with an accelerating or decelerating pattern, some seem isolated without a normal train around. The characteristics of the clicks are similar in both cases, unlike what was found by Martin et al. (2018). Unlike other click trains, we found no superimposed buzzes, which seems to indicate that this type of sound is not emitted by two animals at the same time. Despite some variability, possibly due to a variable signal to noise ratio, a general pattern of a larger bandwidth and a lower intensity is visible for most of the clicks with short ICI, as shown in Fig. 6, confirming Götz, Antunes & Heinrich (2010) measures. No visual follow-up was done, so that we cannot link the buzz to a specific behavior.

Feasibility of medium-term monitoring

Even though the experiment described in this study only lasted one week, we classified it as medium-term monitoring because it combined characteristics of the two usual ways of studying acoustic productions of coastal odontocetes: several months of long term monitoring by mean of detectors versus few hours short term studies with dipping hydrophones from a boat. We think that our approach could be a good alternative for future studies.

Long term monitoring, such as few months of recording at a sample rate of around 500 kHz is still not feasible in remote areas or without very large resources. It produces about one terabyte of data in a period of ten days, which is the order of magnitude of the duration of our experiment. The alternative of a very low duty cycle is not very well adapted to odontocetes which produce a few minutes of sound at each passage as presented in this work. On the other hand, the short term studies are possibly more invasive. Much fewer clicks are recorded and the whole repertoire of the recorded species is difficult to obtain. Our protocol enables to have a relatively non invasive experiment along with a detailed audio data set which is quasi continuous for several days. The presence of the material probably did not modify, in the medium term, the acoustic behavior of the odontocetes during the recordings. Nevertheless it is worth noting that during the maintenance, a group of Chilean dolphins present in the zone went away while two dolphins of the group stayed and repeatedly approached the diver. Afterwards, no acoustic production was recorded by HQB nor detected by the CPOD during three days. The setting-up of this type of device and/or the unusual presence of a diver could have had an impact on the medium-term presence of coastal odontocetes. Obviously, we cannot state that our interactions with the Chilean dolphins were the reason why clicks were not detected afterwards, however, as a conservative measure, we would recommend to install, maintain and retrieve the instruments when odontocetes are not present.

Finally, we also showed the feasibility of acoustic monitoring of NBHF species in remote habitat, with university built material. Our device is adapted to simple installation (two stable feet) in the sheltered channels of Patagonia, at low depth but can be modulated to other uses, depending of the place or species to monitor. Medium-term monitoring with full time recording could also offer unique opportunities to study species occurrence and behavior in the context of anthropogenic activities (with noise signatures, e.g., boats, salmon farming activities).

Conclusion

Medium-term recording shows an interesting complementarity with other more traditional methods of acoustic studies of small dolphins or porpoises in remote areas. They allow an insight on a repertoire much more diverse than was previously considered. This detailed examination of clicks recorded from animals as little disturbed as possible opens new questions concerning sound production or sonar utilization by these species. To complete this work, we suggest medium-term studies should be associated with visual monitoring, ideally from the shore, to avoid disturbing the animals, and taking advantage of the very coastal habits of these species in remote areas. On the other hand, by comparing our detection results with C-POD detection, this study also validates the use of standard detectors for large term monitoring of the presence of small cetaceans in remote areas.

Working with local communities and international universities, affordable missions can be designed to know more about these sensitive species, very prone to be affected by the unregulated development of human activities on the coastal environment.

Supplemental Information

Supplemental Information 1 Code for click detection, written in Octave, for dolphin click detection

Click here for additional data file.

Supplemental Information 2 Code for click parameters extraction, written in Octave

Click here for additional data file.

We thank “Agrupación de turismo naútico y conservación de cetáceos de Puerto Cisnes” and particularly Celestino Ancamil, Israel Ancamil, Francisca Castro Muñoz, Jorge Duamante, Catalina Paz Jorquera, Cristian Maldonado y Hector Pérez Baez for their knowledge of the studied zone and their participation in field work. The authors want to thank CIEP staff: Daniel Pérez and Claudio Herranz for the preparation of the devices and the logistics.

Additional Information and Declarations

Competing Interests

Author Contributions

Data Availability

The authors declare there are no competing interests.

Julie Patris conceived and designed the experiments, performed the experiments, analyzed the data, prepared figures and/or tables, authored or reviewed drafts of the article, and approved the final draft.

Franck Malige conceived and designed the experiments, performed the experiments, analyzed the data, prepared figures and/or tables, authored or reviewed drafts of the article, and approved the final draft.

Madeleine Hamame conceived and designed the experiments, performed the experiments, prepared figures and/or tables, authored or reviewed drafts of the article, and approved the final draft.

Hervé Glotin conceived and designed the experiments, prepared figures and/or tables, authored or reviewed drafts of the article, and approved the final draft.

Valentin Barchasz conceived and designed the experiments, prepared figures and/or tables, authored or reviewed drafts of the article, and approved the final draft.

Valentin Gies conceived and designed the experiments, prepared figures and/or tables, and approved the final draft.

Sebastian Marzetti conceived and designed the experiments, prepared figures and/or tables, and approved the final draft.

Susannah Buchan conceived and designed the experiments, authored or reviewed drafts of the article, and approved the final draft.

The following information was supplied regarding data availability:

The codes of click detector and of parameters extraction in Octave are available in the Supplemental Files.

The raw acoustic data is available at Zenodo: PATRIS Julie. (2023). Soundscape and small cetacean sounds. Puerto Cisnes May 2021. [Data set]. Zenodo. https://doi.org/10.5281/zenodo.7708906.

The files are named after the UTC date of recording.

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
