# Peer review of "Medium-term acoustic monitoring of small cetaceans in Patagonia, Chile"

_PeerJ, doi:10.7717/peerj.15292_

## Round 0.1 · original submission · Major Revisions

Please follow the comments made by the Reviewers, I am sure it will improve your paper. Please include and check references, the spelling of the species, justified your assumptions, improve your English. Especially reviewers 1 and 3 have made a huge effort in correction and suggestions, please take them into account that will improve the paper.

·

Basic reporting

The overall article is well structured (but see below). It could do with a fluent English speaker reading it. For instance, "sensible" is used instead of "sensitive" and "mid-term" is not the right word to express the intended, which I suspect is is "medium term".

Graphs are generally fine and sufficient info is given for the spectrograms etc. BUT proper units must be supplied instead of normalized values! This will make the whole thing much more useful to the reader.

Space can be utilized better in the figures. Calibration values of both the preamplifier, the hydrophone and clip level of the AD must be given.

On the one hand, too much detail goes into describing for example how frequency centroid is calculated (which is known and previously published), on the other, the fact that frequencies outside the range 100-200 kHz are not part of the calculation remains to be found out by looking at the script. Good that the script is there. Many of the variable names are French and the script must be worked over in terms of efficiency. Preallocate the output vectors (and delete empty entries after the loop).

Experimental design

In terms of reproducible design, this is fine. But more data is needed on the hydrophone and preamplifier and perhaps the type of low-pass that was used inside the sampler.

I am uncertain of the extent to which a clear hypothesis was formulated before the experiment ran, but I do think ONE aspect that was intended was the comparison between true time series evaluation and the C-POD data, and I believe this is an important thing to test. Therefore focus should stay on this aspect.

Validity of the findings

As stated above I found the comparison with CPOD data and a time series collected over medium duration deployment super interesting and relevant. I think this part is super good and important to a lot of people using PAM, because the CPODs are very widely used. Good job.

The discussion about outlier signals and the comparison to other I believe to be superfluous and possibly based on insufficient control and knowledge over many aspects of the nature of the data collected. If you do insist on pursuing this, you should limit yourselves to data that both the C-POD and the QHB have identified as belonging to NBHF species. The discussion as it is now is solely built on an observed multimodal distribution of peak frequencies of ALL data without any other factors considered, such as ICI and SNR and received level, which are very important in determining here the exact peak in a spectrum will occurs. The centroid is a much better measure, and behold: the distribution of that parameter is completely smooth and in agreement with Götz et al.

You should also allow peak frequencies to be found below 100 kHz. Extend the frequency vector, also for the sake of the frequency centroid. And state in the methods section what the limits were.

Additional comments

These are comments I jotted down when reading it the first time round. Some of the comments overlap with what I’ve stated above.


L54 & 79: few -> little

87-90: This part I find a little disturbing. It’s like you admit that this is probably a better method, but then you have to say why you didn’t use that. I would not necessarily object to arguments along the line of the pricing of SoundTraps: that you wanted a less exclusive study that could be employed without great costs.

Peale dolphin – Peale’s dolphin

141-142 Morisaka & Connor 2007 would have to be mentioned here as well.

158 To justify this, the SNR of the preamp must also be above 130 dB, which I’ve never encountered. I don’t think you need more than 16 bits. Ever. But you should check that. You can see if you other results with a data reduced to 16 bits!

168: “calibrated with a flat response up to 150 kHz (no available calibration beyond)”. But this makes your criticism of the low-pass filter settings in cPODs and (older) SoundTraps a bit odd, doesn’t it? You need more information about the sensitivity etc.

184: “excluding any other species of cetacean”. But you said in the introduction that the other species were elusive. How can you be *sure* the different call types did not belong to any of the other species hanging around?

209 – 210 That’s a very complicated way to say mean frequency. And what’s with the that ï in centroid??

207 – 226: This list can be made much less verbose by referencing papers using these. It’s not really something you invented, so if you feel, they can be put in supplementary. Perhaps as the Octave/Matlab code used.

212 unfrequent -> infrequent

216 – 226 here you fail to describe if your calculations use the last sample above or the first sample below as seen from the peak.

245-246 This is interesting!

247 & 296 – sensible? You mean sensitive, I’m sure.

Fig 5. Use the same number of bins so you can use the same y-axis for all subplots

310: In a whole -> Overall

311: is that peak OR centroid or peak AND centroid?

318: “Thus, most of the detected clicks can be practically considered on-axis” That’s just not correct. You don’t know the distance to the animals, and closer off-axis calls could easily be 10 dB higher than on-axis calls from farther away.

351: “Soundtrap (low-pass filter at 150 kHz).” This is not true for newer versions of the Soundtraps. But they were never used on this species either ...

Fig 6: Your peak frequency search (and frequency centroid vector) start at 99 kHz, but the example spectrum in Fig 6 lower right subplot shows a signal with peak frequency at 100 kHz. This strongly suggests that there could be signals with lower peak frequencies in the data set. In fact, the freq. centroid of that signal must be lower than measured, since some energy is clearly below 100 kHz and therefore doesn’t get included in the estimate.
The spectra should have spectral level (dB re. 1µPa / Hz) as received by the recorder, and the time domain subplots should have proper acoustic units (kPa) on the y-axis.

Reviewer 2 ·

Basic reporting

The document is well written and structured.

Experimental design

I woulds suggest that authors include references to Ocean basins in figure 3, and I think it would be worth to share more information on the mooring design, might be worth to share an illustration of that.

Validity of the findings

I see the value of this work in providing evidence and encouraging the use of autonomous acoustic recordings to study NBHF species, despite the relatively short duration of the recording sessions. I strongly agree on the importance of analyzing larger data sets, specially in NBHF species which have been proven to have the capability of producing a much more diverse acoustic repertoire than previously thought. I encourage authors to combine this mid-term acoustic monitoring with visual monitoring in order to get a better insight into the behavioral context of the various clicks recorded.

Additional comments

One thing was not clear to me along the document. How can authors be sure that the recorded clicks were produced by Chilean dolphins and not by one of the other NBHF species mentioned to inhabit Chilean waters? Is this particular area inhabit only by Chilean dolphins? If that is the case I suggest to explicetely mention that on the paper.

·

Basic reporting

This study reports preliminary findings using a custom-built recorder - it shows both the functionality but also the application of the new recording set up. This part seems strong and quite well presented. The biological inference however is less well supported - should the authors wish to focus on the biological implications then more careful analysis will be needed.
The writing and use of the English language could be improved to ensure that your text is clearly understood by an international audience and that the meaning is clearly conveyed as you intended it. You use “mid-term” but a better expression perhaps would be “medium-term” duration (to contrast with short and long term duration of recordings).
The manuscript should follow the layout/ structure typically used in articles (and in PeerJ), so you will need to adjust your headings and subheadings accordingly, e.g. no section numbering, clearer separation into Introduction, Materials and Methods, Results.....
The manuscript could make better use of existing relevant literature – it is rather narrow in focus and should at least acknowledge the approaches to click classification that exist and provide a broader overview of current PAM methods. You also need to consider all NBHF click producing species as potential sources for the recorded sounds and describe their click characteristics accordingly. Some more up to date references should be used, and references overall need to be formatted correctly and consistently, e.g. there’s information missing in the citations, spelling of Latin species names.

Experimental design

This is an interesting observational study that reports preliminary but exciting findings. It could be improved with better structure of the manuscript and a clearer focus on the key messages – at the moment the paper meanders between a technical description of the recording device and its performance in comparison to a commonly used autonomous recorder, and a description of the clicks recorded with some inference about the click-producing species. The study’s objectives/ aims are not stated clearly and should be defined at the end of the Introduction. The current Introduction is mostly focussed on the geographic area and the species, but the results mostly support technical comparisons. I would recommend that the authors focus this paper on the technical aspects and the application of their newly designed recording system, and thus rewrite the paper with this as focal theme. Robust biological inference will need a more extensive analysis of the generated dataset. The data processing appears to be rigorous, but a bit more information is needed on the methods, e.g. binning of detections for comparisons between instruments (see specific comments), accuracy of detection classifications/ false positive rate.

Validity of the findings

One of the key assumptions of the study is that all the recorded clicks are from Chilean dolphins, and the results are interpreted with that focus. This assumption is not justified appropriately and is not justifiable given the recording set up and the current level of analysis. Given the data it would seem more appropriate for the manuscript to focus on discussing the use of the recorder for the detection of NBHF click producing species, and perhaps for characterising natural and anthropogenic noise leels. If you wish to make inference about species-specific sounds then the existing data will need to be analysed in a way that allows for species discrimination and can show robustly that those clicks indeed belong to Chilean dolphins (and not the other two sympatric NBHF click producing species).

Given the study set up and the results presented in the manuscript it would be better to focus on the strength and the usefulness of the proposed methodological approaches as a means to quantify interactions of cetaceans with anthropogenic activities and their exposure to these activities. TIt would be better to avoid judgemental statements or use very limited anecdotal evidence to make strong statements about the animals’ behaviour (e.g. your interpretation of the dolphins' purported absence after the instrument deployment activities)

Additional comments

Specific comments on “Mid-term acoustic monitoring of Patagonian coastal dolphins”

Abstract:
Overall, the abstract needs some restructuring and clearer focus on the main results and message of your study. The reference to Commerson’s dolphins seems out of place and not needed. It would be better to focus on the main results of your study, with more detail given on the number of detections and patterns in the detections, and the use of the medium-term recorders to address particular research questions.
l.22 spelling: the correct common species name is Peale’s dolphin – this applies to the entire text. (not Peale dolphin)
l.23 spelling: the correct common species name is Burmeister’s porpoise - this applies to the entire text. (not Burmeister porpoise)
l.23 suggest you delete “pristine”: unfortunately Chilean Patagonia, and especially Canal Puyuhuapi is anything but pristine. Human activities, including salmon farming, have been intense and ongoing for decades.
l.25 – delete “fragile” – none of the species is fragile
l.28 – perhaps introduce the use of PAM before discussing storage problems to improve flow/ context of this statement.
l.31-32: needs rewording. The medium-term monitoring you trial is not really an alternative, but an option to respond to different research questions. Real-time recordings allow for species identification. Long-term recordings allow for temporal patterns to be investigated. Your medium-term monitoring offers shorter-term high resolution information, and is the best currently available approach to investigate cetacean occurrence and behaviour in relation to anthropogenic activities.
l. 36-37: you have not shown and cannot assume that the clicks you recorded are from Chilean dolphins. You introduce the 3 NBHF species in the area in the first sentence, and you cannot distinguish the species based on the recorded clicks alone. This applies to the entire manuscript, so please reword accordingly. This limitation does not distract from the value of your work.
l. 39: past tense – were present.
l.40-41: is not very clear. Do you mean that you obtained a large number of clicks that included both on and off axis clicks so that the distortion of the latter will influence the parameter values obtained?
l. 42-43: not “parent”, but a congener (species in the same genus), needs changing also in other parts of the manuscript
l.46-47: this sentence needs clarification. Why would one need to put two recording devices? This seems costly and complex. Two more obvious conclusions would seem that: a) CPODs are quite good but might miss detections and are of limited use for the study of the co-occurrence of cetacean – noise-based anthropogenic activities, b) that acoustic classifiers are needed that also allow for identification of NBHF clicks and ideally also discrimination between NBHF species.

Introduction:
l. 54 – endemic is the wrong term here. None of the cetacean species are endemic to Patagonia, and only Chilean dolphins are endemic to Chile, but your introduction should refer to all NBHF species.
l. 56 “submitted to the contingencies” – incorrect English phrasing. Perhaps subject to/ affected by?
L.56 – should read: locally available equipment
l. 59 – not “coastal dolphins” – the high sampling rate applies to all NBHF species (this includes offshore species) and perhaps odontocetes more widely
l.61 – “the very few studies” – please check that this statement is correct – Soundtraps and other full bandwidth recorders have been around for a while and are now widely used. Surely there must be many more published studies out there.
l. 61 wording – perhaps “autonomous full bandwidth recording”?
l. 69 replace “registered behaviours” with behavioural observations
l.76-77 needs better wording – CPODs for example log certain click characteristics (e.g. wave form) and use these to classify sounds into pre-determined categories
l.88 becoming available (not getting)
l.94 needs rewording – custom-build recorders might be useful for those who have the means to build them; this does not need to be limited or linked to a University. If you have a particular University in mind here then please state this.
l. 97-98 – this sentence needs better wording. It also seems a bit strong to refer to recordings from a boat as “invasive” – this applies to the entire text.
The section headings are not following the journal guidelines. The first section of the main text is the Introduction followed by Materials and Methods. Your current section 1 is part of the Introduction. Your current section 1.1. doesn’t seem very relevant to the paper – you are not evaluating the ecology of cetaceans or the threats to the Chilean fjords or the species that inhabit that area. A short description of the actual characteristics of the deployment site as the first section in the Materials and Methods section would suffice.
l.115-146 – this text could be shortened substantially and should be more focussed on the information that actually matters for your study, i.e. the acoustic characteristics of the species that occur in your study area (Burmeister’s porpoises, Chilean dolphins, Peale’s dolphins, bottlenose dolphins, killer whales) and that your recording equipment is targeted at. There are a number of observational studies published that describe the occurrence of those species in the area (e.g. see Zamorano-Abramson et al. 2010, already in your reference list). The unifying features of Burmeister’s porpoises, Peale’s dolphins and Chilean dolphins is that they are NBHF click producing species, and those click characteristics make them distinguishable from bottlenose dolphins and killer whales. It would be useful if you could briefly explain this. It would also be helpful to set out how large acoustic datasets are usually processed to derive at species occurrence (detections) and species identification. The information on threats to those species in other areas is not really relevant for your study. Focus on why those species are not well studied, how PAM can be used and how it could help better understand their distribution and interaction with anthropogenic structures. Use examples from the ample literature on other species with similar characteristics (e.g. harbour/ Dall’s porpoises) to illustrate the use of those methods.
L.126-127 – you are using outdated IUCN Red List information, please check the current Red List for the correct status and citation for Peale’s dolphins (Least Concern) and Burmeister’s porpoises (Near Threatened).
L. 129-130- your list of threats is rather generic. It would be better to tailor this to the study area and species within, or delete completely.
L. 132-133 – “little boats” – not good English, and also not correct. Peale’s dolphins are known to bow ride on all sizes of vessels that overlap with their distribution. It’s just that small vessels are far more common in the waters where Peale’s dolphins occur.
l. 138-146 text needs rewording. NBHF clicks are not common cetaceans in coastal areas. Only a small number of species produce these click types. Chile is unique in that three NBHF click producing species overlap substantially in their ranges. NBHF clicks are similar between species that produce them, but there are also differences in peak frequency between the porpoises and delphinids. You might also want to cite some of the more recent works describing the relaxation of acoustic crypsis in Cephalorhynchus (e.g. the works of Martin et al., 2021, 2019, 2018)
l. 141 needs plural – “that do not hear above....”
l. 142-143: not only Chilean dolphins produce buzzes – all NBHF species do. Martin et al., 2019 also do not state this about Chilean dolphins. This sentence needs better wording.
At the end of the Introduction you should state clearly your study’s objectives/ aims. These are missing, and some of the aims only become clear in the discussion.

Methods
l. 150 “this state of the art recorder has (not have)”
Figures 1 and 2 – these are rather technical – if the focus of the manuscript is shifted to the technical aspects/ descriptions then these would seem useful to include in the main text. With the current manuscript these seem to be somewhat mismatched with the application/ biology focussed text and as figures are not particularly accessible to a non-engineering focussed reader.
l.183-184 – This statement is not supported by evidence, and the assumption you make is likely not correct. You cannot use your short duration visits or those of tour operators to the site to exclude the occurrence of other species from the deployment site. All three NBHF click producing species are known to use this general area and therefore you cannot assume that the clicks you recorded around the clock and over weeks are only of Chilean dolphins. You will need to use species-specific classifiers to determine this. Your results also indicated that other species might be the source of some of the clicks (see below)
L. 194 – It is laudable that you developed your own methods for detecting the NBHF clicks in the dataset. However, you do not provide much evidence of the accuracy of your approach. Did you look into false positive rate? Into the effects of ambient sounds on the click classification?
L. 203-205 - Please check the published studies on Peale’s dolphins and Burmeister’s porpoises for the parameter values derived for those species. As above, you cannot assume that all recorded clicks are from Chilean dolphins. Your results also suggest that this might not be the case.
l. 209 - Spelling: centroid
l. 221 – wording- what’s “probability density”?
l. 229 – This should be under the main section heading called Results.
L. 230-231 – It’s commendable that you state clearly that there were issues with the recorders. It would help if you could also explain what the issues were when the instrument malfunctioned, and why you consider that the recorder worked well during the other periods, and you therefore have confidence in the quality of the recordings.
l. 233 - this statement is not clear, what’s “Go”?
l. 235-238 – Some of these sentences belong in Materials and Methods, not Results. Can you provide some references or a better explanation of why you chose that definition for “events”? How does your definition of events (click trains separated by less than 20 min) and the separation of events (30 min or more) align with the term “detections per 10 -min” in Figure 4? Where do you show your detected events and how do these align with the definition for detections in Figure 4?
Figure 4 – this is the main data figure supporting the comparison between CPODs and QHB recorder. The figure quality and presentation could be improved. The green line presumably depicts the tidal amplitude but this is not explained. Also, the green colour is not particularly visible (green and red are also not great colour combinations for disability access). The bars could also be bigger with a more intervals on the x-axis and bigger labels.
L.246-247 – This sentence needs better wording. It is not clear what you mean, and why are the results from the QHB recorder deemed more “sensible”? Did you do any sensitivity analysis of your definition of clicks? Any investigation into false positives? It would be great if you could show some quantitative results here.
l.248-249 – what are “long duration motors”? Do you mean engine noise that remained static (as in same level of intensity indicating no movement)? How often did this occur? Could any of the background noise have affected your detection of NBHF clicks in the data? Please clarify.
l. 251-155 – this section seems rather contrived and setting you up for an unsupported over-interpretation. You state above that you did not investigate background noise (which could affect substantially the detection of clicks and thus affect your inference of dolphin occurrence). It would be good if you could provide a more broadband characterisation of the sounds recorded. It is not clear why boats/ divers would really affect the dolphins’ use of the site (as you discuss below), and as far as I’m aware there are no studies that have reported anything similar for any other species, including during the use of C-PODs with divers with Chilean dolphins (unpublished thesis of the University Austral, Valdivia, D. Filun).
Table 1 could make better use of space, i.e. represent the click parameters in rows, and also add the values from the existing literature for the relevant NBHF species for comparison. This would better support your statement in l. 257-258.
l.262-263 – needs better wording.
The entire section points to issues with your interpretation of all clicks belonging to Chilean dolphins. Burmeister’s porpoises tend to have peak frequencies of about 10 kHz higher than either Chilean or Peale’s dolphins. You cite some of the relevant literature but do not seem to really acknowledge the implications of your findings, particularly with regard to the validity of your assumptions that all clicks are from Chilean dolphins. It would be useful to compare the recorded click characteristics with all three NBHF species and what is known about them.
Discussion
L.272-273 – wording: better to state- “had a larger bandwidth”
L.285-308 – some good ideas are presented in this section. The wording however needs improving for correct English grammar and better clarity (in many of the sentences, e.g. l.307-308). You also need to refer to NBHF click producing species, and not specifically Chilean dolphins (see previous comments). In addition, some of the information in this section should be given in other sections – for example the justification for the choice of event window (Methods) or the differences between QHB and CPOD detection (Results). It’s good to see you consider the sensitivity of the different instruments, but you provide no calibration or checks for false positive rate for your QHB recorders. It is reassuring to see that almost all “events” match between the different recording instruments, but that alone does not reveal which instrument is better or more accurate. It is well understood that individual clicks are not a useful measure (also because clicks within a train are clearly correlated, and multiple clicks or click trains in short succession are also not independent of each other, so do not form a useful unit for statistical analysis). Your section heading and concluding sentence might arise from the results but you did not set out to validate CPOD performance. So perhaps if this is one of the objectives then this should be introduced more clearly in the introduction. It also would seem more appropriate to consider the well established CPODs as means of validation of your novel instrumental set-up, rather than the other way round. It is reassuring that both sets of instruments broadly find the same patterns, though one could also argue that you categorized your data (i.e. events) in a way that gives rise to this matching pattern.
l.310- 333: the same comments about species discrimination, or lack thereof, apply here. You do not know that all your recorded clicks are all from Chilean dolphins (in fact your results seem to point at different species, e.g porpoise clicks would match well the higher peak at 135 kHz), and you should consider all three NBHF clicking species as potential sources in your recordings. You also need to pay more attention to the definition of clicks used in other studies. Most other studies aim to consider only on-axis clicks, and for comparisons usually use the loudest clicks in a click train. You seem to be using all clicks; this poses two potential problems: subsequent clicks are not independent (e.g. issues of pseudo-replication) and including off-axis clicks with distorted signals given the high level of directionality of NBHF clicks will influence your parameter measurements. While off-axis click are useful to consider (given that this reflects the reality of PAM) they also affect the direct comparison with results from other studies that use on-axis clicks. A more nuanced/critical discussion of your click classification process would be useful here.
L.334-341: see above comment on differences in methods and comparisons that might play a role for explaining the results. The presence of a boat is a very unlikely factor for changing the click characteristics, especially with the engine off (as done for the recordings used by Rojas-Mena or Goetz et al). Also, while geographical differences might exist, this is rather unlikely to be the case to the extent as you make out (e.g. compare the similarities of NBHF clicks across the Cephalorhynchus). Southern Chiloe (where previous recordings were done) is less than 300 km from your study site (not 1,000 km as you state).
L. 342-351; some good points presented here but the wording needs improving for better clarity (e.g. correct phrasing, perhaps someone could proof-read the paper with a more critical eye on English grammar and language?)
l.360-365: some more careful thought is needed here. You cannot infer how many individuals were involved in situations where buzzes occur. Only one individual might be vocalising at the time (i.e. non-overlapping click trains) to avoid masking or because the other dolphins in the group are eavesdropping. Chilean dolphins occur in groups both during foraging and socialising, as do Peale’s dolphins. Burmeister’s porpoises tend to be more solitary. As you state, you have no information on behavioural context (and also not about species ID). So perhaps the discussion here would be better directed at the use of PAM and the limitations of your current setup than speculation about the behavioural context of buzzes.
L.367-389: This section is rather weakly articulated and seems to miss the main strong points of your study. I don’t think you can argue that your approach is better than short-term or long-term monitoring as you show no such results. The length of the recording period (with or without observers present) is more relevant to consider in the context of the scientific questions one wishes to answer. Clearly, recording dolphins in situ while ascertaining species ID is crucial for sound characterisations of a specific species, acoustic species identification and discrimination. Your method presents no alternative to that (and studies like yours which generate a large volume of broadband acoustic recordings would benefit from automated species identification approaches). The long-term monitoring using CPODs is also relevant and not replaced by your study approach as it allows temporal patterns in occurrence to be investigated (e.g. diel, seasonal patterns). Your medium-term monitoring with full-bandwith recording equipment offers unique opportunities to study species occurrence (based on acoustic detections) in the context of anthropogenic activities (with noise signatures, e.g. boats, salmon farming activities). Neither of the other two duration approaches can currently offer such a comparison because of recording limitations. It seems strange that you do not at all discuss this point which seems to me to be the main benefit of your recording setup.
L.376 (and in other places): this might be an English language issue but it seems disproportionate to call boat-based acoustic recordings “invasive”.
l.383-389: the inference in this paragraph is entirely anecdotal conjecture and should have no place in a scientific study. There are many alternative explanations why dolphins might not have been recorded after instrument deployment/ servicing. I’m not aware of any evidence in other places or other species that the presence of a diver (and/or a boat) caused avoidance of an area, especially not over such a long time period, and it is very unlikely that this was the cause for the differences in your recordings. I would strongly suggest you remove this section from the manuscript (or providesupporting evidence for your points).

Supporting documents
Image file entitled “View” – this picture seems to bear no relation to the study (it shows a hill with lovely trees but no sea at all). What’s the purpose of this image?

References:
Please check the spelling of the species’ Latin names in all citations. Genus names should be in capital letters. Latin names should always be in italics.
Some references are also incomplete with missing journal names or missing page numbers.
IUCN Red list status for Peale’s dolphins and Burmeister’s porpoises were updated in 2019, you refer to the previous listings, please update the references.

---

## Round 0.2 · Major Revisions

The reviewers still make important corrections to take into account before the publication of the paper. you should carefully follow your suggestions and send your paper for grammar checking to an English-speaking colleague. I hope to have the paper with the corrections soon, so that I can accept it after hearing from you.

·

Basic reporting

This is for a special issue, so It's probably OK

Experimental design

OK

Validity of the findings

OK

Additional comments

L220: “The envelope is obtained from the Hilbert transform of 1ms of signal around the clicks”. This is all Mathworks fault. It’s called the analytical signal. Here you can just write: “The envelope is obtained using the absolute values of the analytical signal (Hilbert transform in Matlab) corresponding to 1ms of real-valued signal around the peak of the clicks”.

On Fig. 6: “spectrogram of the signal with a FFT on 2^10 points except for the right picture (2^7 points), Blackman window, 50% overlap.“ The important parameter is the chunk length, which should always be stated. For odontocete clicks, a short chunk (like 30-40 µs) is appropriate. For longer segments with several clicks, it CAN be good with a longer chunk to give less freq. smear a more time smear (to make it visible). Give the FFT size as e.g. 1024.

And do limit the dynamic range of the plot.

If you use a modern Matlab’s spectrogram, this could be:
s=signal;
sr=512;
N=round(0.04*sr);
scr=[1024,2048]; %screen/image resolution here
step=max([round((length(s) - N)/scr(2)),1]);
spectrogram(s,blackman(N),N-step,max([N,scr(1)*2]),sr,'yaxis'),
caxis(max(caxis)+[-30,0])
colorbar off

With regards to the normalization of values, it is certainly less informative than actual pressure values. I find the response that normalized values are “more legible” flippant. You can adjust the upper y-scale limit to encompass the signal.

For the spectra, I’d allow normalization, but it has to say dB re. max power on the y-label

·

Basic reporting

This is a resubmission of a much improved article. I commend the authors for their great efforts and for having taken onboard many of the reviewers’ previous comments. There are several areas though that still require attention and substantial editorial revision. I’ve provided non-exhaustive comments on the pdf which I hope the authors will find useful in revising this manuscripts.

The English and writing are much improved but there are still many poorly worded passages, grammatical errors and wrong word choices. These detract from ease of understanding of the manuscript and in parts obfuscate the meaning. Perhaps if one of the co-authors is a native speaker they could give the manuscript a bit more of a careful proof-read? At the moment the manuscript does not seem to meet the PeerJ requirement of “Clear, unambiguous, professional English”.
I have added substantial comments/ highlights on the revised pdf to point out some of the main glitches but it’s beyond reviewer remit to provide language assistance. Beyond language there are also some formatting/labelling glitches that need to be addressed.

Title: given that you have Burmeister’s porpoises now mentioned in the abstract perhaps the title should be “Medium-term acoustic monitoring of small cetaceans in Patagonia, Chile”?

There is still some confusion and inconsistency in the taxonomic references used throughout the manuscript. It would seem preferably to settle on small cetaceans/ small odontocetes throughout and avoid referring to “dolphins” (this ignores the porpoises) or “marine mammals” which is far too broad a polyphyletic category.

Experimental design

The aims of the study are still not clear and have not been stated. Please define your research questions more clearly and articulate how your research fills the identified knowledge gaps. For example, the subsection headings in the discussion give some indication of what you might be trying to achieve.

Calling your work an “experiment” is confusing – what is the actual experiment? Testing the different recorders? Perhaps your work could be better described as trialling a new acoustic recording system with the aim to provide more/better tools for PAM studies of small cetaceans in remote locations?

I’m still not convinced by the structure of the introduction. Most of the text on the three potential species and the study site seems redundant and not clearly connected to your work – you don’t really make use of that information throughout the manuscript.

Validity of the findings

The figures are generally good and show the results. They appear rather small and low resolution but this might be only the case in the review manuscript. The figure legends and labelling need to be revised and amended – there are omissions and inconsistency in the labelling of what’s shown. Please also check that figures and tables are referred to using consistent notation (spelling as in the legend/ heading, usually with a capital letter at the start).

As in the previous review I still do not accept your interpretation of the dolphins' avoidance of the area after you installed the instruments. You provide no real evidence for this, this is not based on systematic observation, it's pure conjecture, and it's also somewhat ignorant of the species' normal behaviour which does include fast swimming alongshore. Calling this "fleeing" is naive at best.

The rest of the data interpretation, particularly the acoustic-focussed part are much improved and sound. You provide the underlying data and the conclusions drawn from your acoustic analyses seem robust.

Additional comments

Please see the commented pdf for more detailed comments.

---

## Round 0.3 · accepted · Accept

Congratulations, your paper is ready to be published in PeerJ, you have addressed all the comments and suggestions made by the reviewers and by me.